# AIM: Adapting Image Models for Efficient Video Action Recognition

**Taojiannan Yang**[1][*] **Yi Zhu**[2], **Yusheng Xie**[2], **Aston Zhang**[2], **Chen Chen**[1], **Mu Li**[2]
[1]Center for Research in Computer Vision, University of Central Florida  [2]Amazon Web Services
taoyang1122@knights.ucf.edu  yzaws@amazon.com  chen.chen@crcv.ucf.edu

## Abstract

Recent vision transformer based video models mostly follow the "*image pre-training then finetuning*" paradigm and have achieved great success on multiple video benchmarks. However, fully finetuning such a video model could be computationally expensive and unnecessary, given the pre-trained image transformer models have demonstrated exceptional transferability. In this work, we propose a novel method to Adapt pre-trained Image Models (AIM) for efficient video understanding. By freezing the pre-trained image model and adding a few lightweight Adapters, we introduce spatial adaptation, temporal adaptation and joint adaptation to gradually equip an image model with spatiotemporal reasoning capability. We show that our proposed AIM can achieve competitive or even better performance than prior arts with substantially fewer tunable parameters on four video action recognition benchmarks. Thanks to its simplicity, our method is also generally applicable to different image pre-trained models, which has the potential to leverage more powerful image foundation models in the future. The project webpage is https://adapt-image-models.github.io/.

## 1 Introduction

The "pre-training then finetuning" paradigm has played an important role in computer vision. The key to this paradigm is a well pre-trained image model, which can provide strong transferability to downstream tasks through finetuning. Recently, large foundation models (Radford et al., 2021; Yuan et al., 2021b; Tong et al., 2022; Jia et al., 2021; Wang et al., 2022b) can even demonstrate remarkable few-/zero-shot performance given their learned superior visual representations.

In video understanding, a common practice is also bootstrapping from an image pre-trained model and then finetuning on the video data. There are two dominating directions as shown in Fig. 1, one is to extend an image model with additional temporal module (Lin et al., 2019; Zhu et al., 2019; Arnab et al., 2021), the other is to inflate an image model to a video model (Carreira & Zisserman, 2017; Liu et al., 2022). However, there exists at least two drawbacks for the aforementioned methods. First, most approaches require full finetuning (*i.e.*, updating all the model parameters during training) to achieve promising results on common video benchmarks. This is quite costly in terms of both computation and memory footprint, *e.g.*, 1200 Tesla V100 GPU hours to train Liu et al. (2022). Second, it also remains questionable that whether it is necessary to fully finetune pre-trained image models given that they have demonstrated excellent transferability. An inadequate finetuning on downstream data might destroy the well generalized representations from such foundation models.

To overcome the drawbacks, a research direction termed parameter-efficient transfer learning has been trending in natural language processing (NLP) (Houlsby et al., 2019; Lester et al., 2021; Ben Zaken et al., 2022; Hu et al., 2022). The goal is to only finetune a small number of (extra) parameters while keeping large pre-trained language models (Devlin et al., 2018; Brown et al., 2020) frozen to attain strong performance. With the rise of large vision transformer (ViT) models, such techniques have been recently introduced to computer vision for efficient transfer learning. However, existing works either focus on tuning a pre-trained image model for image tasks (image-to-image) (Bahng et al., 2022; Jie & Deng, 2022; Jia et al., 2022), or tuning a pre-trained video model for video

---

[*]Work done during an internship at Amazon Web Services.

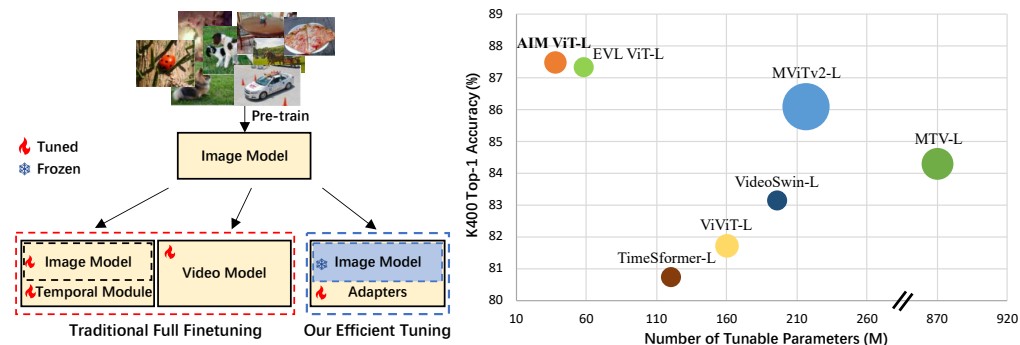

Figure 1: **Left**: Pipeline comparison between traditional full finetuning and our efficient finetuning. **Right**: Performance comparison on K400 dataset (Kay et al., 2017). Bubble size indicates GFLOPS at inference time. Our proposed AIM achieves the highest accuracy while enjoying significantly less number of tunable parameters and GFLOPS.

tasks (video-to-video) Chen et al. (2022). Directly leveraging pre-trained image models for efficient transfer learning to video tasks (image-to-video) is less explored, because image models lack the capability of temporal reasoning.

In this work, we introduce a new way to **A**dapt pre-trained **I**mage transformer **M**odels (AIM) for efficient video action recognition. By freezing the pre-trained image model and adding a few lightweight adapters (Houlsby et al., 2019) during finetuning, we show that our proposed AIM can achieve competitive or even better results than previous state-of-the-art methods with substantially fewer tunable parameters (Fig. 1 right). To be specific, we first introduce adapter after self-attention layer in a transformer block to perform *spatial adaptation*. We show that a well pre-trained image model is sufficiently good for spatial modeling in video understanding. Then for temporal modeling, we simply reuse the image pre-trained self-attention layer but apply it to the temporal dimension of video input, forcing it to model the relationship across different frames. An adapter is also appended for *temporal adaptation*. Finally, we perform *joint adaptation* by adding another adapter in parallel to the MLP layer in a transformer block. To summarize, we make the following contributions:

1. We propose a new way to adapt pre-trained image transformer models for efficient video understanding. Our method is generally applicable to different image pre-trained models, simple to implement, and cost-effective to train.

2. Our method is significantly more efficient than fully finetuning a video model, *e.g.*, on Swin-B backbone, we can reduce the memory footprint by 50% and training time by 42% compared to VideoSwin (Liu et al., 2022).

3. AIM achieves comparable or higher performance than previous fully finetuned state-of-the-arts on 4 video action recognition benchmarks, *e.g.*, 87.5% on K400 with 38M tunable parameters.

4. Our method also brings data efficiency, *e.g.*, AIM outperforms counterpart TimeSformer (Bertasius et al., 2021) by 9% absolute accuracy improvement when using 1% of the training data.

## 2 RELATED WORK

**Image pre-trained models.** ViT (Dosovitskiy et al., 2020) and its variants (Liu et al., 2021; Wang et al., 2021b; Yuan et al., 2021a; Dong et al., 2022) have been proposed to achieve state-of-the-art performance on image recognition. Once trained, these models could also serve as good initialization for transfer learning to downstream tasks. In terms of training techniques, they are commonly trained on large-scale labeled datasests (Deng et al., 2009; Sun et al., 2017; Zhai et al., 2022) in a supervised manner. To alleviate the labeling cost, self-supervised learning methods (Chen et al., 2021; Bao et al., 2021; Zhou et al., 2021; He et al., 2022b; Xie et al., 2022) are introduced to learn effective representations from unlabeled data. Recent works (Radford et al., 2021; Jia et al., 2021; Yuan et al., 2021b; Wang et al., 2022b) adopt large-scale multimodal data (*e.g.*, image-text pairs) for model training, which leads to even more powerful visual representations. In this work, thanks to the simplicity of our proposed method, we could take advantage of these well pre-trained image models and adapt them efficiently to solve video tasks.

**Video action recognition.** A paradigm shift from using convolutional networks (Carreira & Zisserman, 2017; Tran et al., 2018; Yang et al., 2021; Lin et al., 2019; Feichtenhofer et al., 2019) to transformers has been observed for video action recognition. Most works use image pre-trained models as initialization and extend them to video models by introducing new temporal modules (Bertasius et al., 2021; Arnab et al., 2021; Zhang et al., 2021b; Yan et al., 2022) or inflating them to video models (Liu et al., 2022). Another direction is to directly pre-train a video model in a self-supervised manner (Kuang et al., 2021; Feichtenhofer et al., 2022; Zolfaghari et al., 2021; Tan et al., 2021). However, all these models are fully finetuned on video data, which makes the training cost unaffordable to most researchers and practitioners. There are some recent works Ni et al. (2022); Ju et al. (2021); Wu et al. (2023; 2022) extending CLIP to perform action recognition, but they are multimodal methods which requires additional text branch. Our proposed AIM leverages existing pre-trained image models (no need for video model pre-training), only tunes a small number of model parameters (much more efficient than full finetuning), and achieves comparable or even better performance than previous state-of-the-arts.

**Parameter-efficient finetuning** techniques (Houlsby et al., 2019; Hu et al., 2022; Lester et al., 2021; Li & Liang, 2021; He et al., 2022a; Ben Zaken et al., 2022; Sung et al., 2021; Qing et al., 2022) are first proposed in NLP since fully finetuning the increasingly larger language models for various downstream tasks becomes less feasible. Their goal is to reduce the number of trainable parameters thus lowering the computation cost, while reaching or surpassing the performance of full finetuning. Recently, parameter-efficient transfer learning is also studied in computer vision (Jia et al., 2022; Bahng et al., 2022; Chen et al., 2022; Jie & Deng, 2022; Gao et al., 2022). All these methods focus on adapting models in the same domain (*e.g.*, image-to-image or video-to-video), while our method adapts an image model for video tasks. One concurrent work (Lin et al., 2022) also studies how to adapt image pre-trained models for video action recognition. However, there are several major differences. First, they add new trainable decoder branches, which consist of 3D convolutions and cross-frame attention, to the frozen image encoder. We simply reuse image pre-trained self-attention to perform temporal modeling, while enjoying better performance and less tunable parameters. Second, our method is shown to be compatible with different image models, while Lin et al. (2022) only shows its effectiveness on CLIP image encoder.

# 3 METHODOLOGY

In this section, we first briefly describe ViT and video baselines (Sec. 3.1). Then we introduce spatial adaptation (Sec. 3.2), temporal adaptation (Sec. 3.3) and joint adaptation (Sec. 3.4), to show how we adapt a pre-trained image model for effective video modeling step-by-step.

## 3.1 PRELIMINARY

After the seminal work of Vision Transformer (ViT) (Dosovitskiy et al., 2020), transformer-based models have been widely adopted in various computer vision tasks, including video action recognition. In this work, we focus on adapting pre-trained image transformer models and compare them to fully finetuned video transformer models, unless otherwise stated.

More specifically, ViT handles an image as a sequence of small patches. Given input image $\boldsymbol{x} \in \mathbb{R}^{H \times W \times C}$, ViT first splits the image to $N$ non-overlapping patches and maps each patch to a $D$-dim patch embedding via a trainable linear projection (Qian et al., 2021; 2022). Here, $(H, W)$ is the image resolution and $C$ is the number of channels. Patch embeddings $\boldsymbol{x}_p \in \mathbb{R}^{N \times D}$, where $N = HW/P^2$ and $P$ denotes the patch size. Then a learnable [class] token is prepended to $\boldsymbol{x}_p$ as $\boldsymbol{x}_0 = [\boldsymbol{x}_{class}; \boldsymbol{x}_p] \in \mathbb{R}^{(N+1) \times D}$. To encode positional information, positional embeddings $\boldsymbol{E}_{pos} \in \mathbb{R}^{(N+1) \times D}$ are added to $\boldsymbol{x}_0$ as $\boldsymbol{z}_0 = \boldsymbol{x}_0 + \boldsymbol{E}_{pos}$, where $\boldsymbol{z}_0$ is the final input being fed to a sequence of transformer blocks. Each transformer block is composed of a multiheaded self-attention (MSA) and a MLP layer, together with Layernorm (LN) and skip connections, see Fig. 2(b). The computation of a standard transformer block can be written as

$$\boldsymbol{z}_l' = \boldsymbol{z}_{l-1} + \text{MSA}(\text{LN}(\boldsymbol{z}_{l-1})), \tag{1}$$

$$\boldsymbol{z}_l = \boldsymbol{z}_l' + \text{MLP}(\text{LN}(\boldsymbol{z}_l')), \tag{2}$$

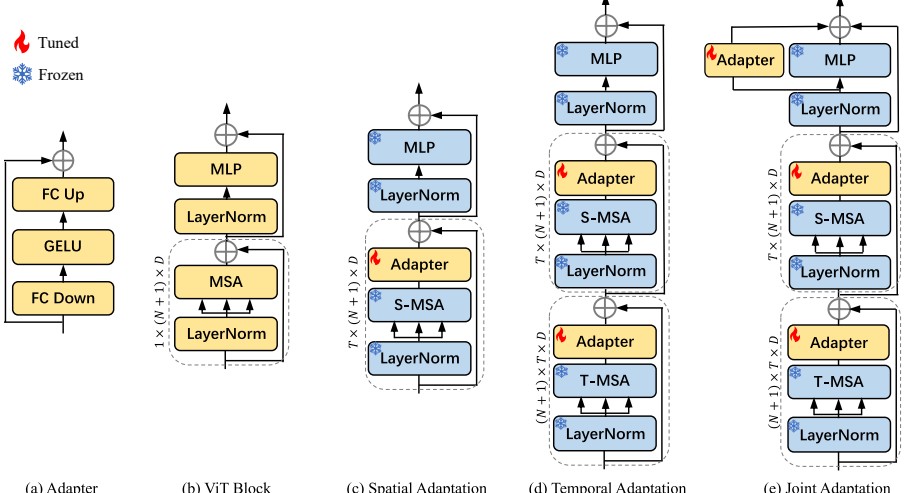

| (a) Adapter | (b) ViT Block | (c) Spatial Adaptation | (d) Temporal Adaptation | (e) Joint Adaptation |

Figure 2: We show how we adapt a standard ViT block (b) for video action recognition, by gradually adding spatial adaptation (c), temporal adaptation (d) and joint adaptation (e). Note that S-MSA and T-MSA share weights but are applied to different input dimensions. During training, only newly added Adapters are updated while all the other layers are frozen.

where $z_{l-1}$ and $z_l$ denotes the input and output of the $l$-th transformer block, respectively. Finally, the learned [class] token $x_{class}$ from the last transformer block is used as global visual representation and fed into a classification head to make the prediction.

**Space-only and space-time models for video**. A video is a stack of frames with temporal structure. Hence, video understanding requires the model to learn both good appearance representations in each frame (spatial modeling) and also infer the temporal information across frames (temporal modeling). In order to leverage an image transformer model for video tasks, one key thing is how to perform temporal modeling. A simple baseline, termed space-only model, process each video frame independently by an image model. Given $x \in \mathbb{R}^{T \times H \times W \times C}$, where $T$ is the number of frames, space-only model will get $T$ [class] tokens where each [class] token stands for the representation of each frame. These $T$ [class] tokens will be averaged as a way of temporal modeling for final prediction. In order to enhance the capability of temporal modeling, recent works (Bertasius et al., 2021; Arnab et al., 2021; Zhang et al., 2021b) introduce space-time model by adding new temporal modules to image models. These models are now the top performers on most video action recognition benchmarks, however, their training costs are prohibitively high due to full finetuning. Given the increasingly larger and more powerful pre-trained image models, in this work, we study how to efficiently adapt them for video action recognition.

### 3.2 SPATIAL ADAPTATION

Since image pre-trained models have been trained on large-scale datasets and demonstrated strong transferability to downstream tasks, we believe they could achieve good spatial modeling in video action recognition with minimal finetuning.

Inspired by efficient finetuning techniques (Houlsby et al., 2019; Lester et al., 2021; Li & Liang, 2021; Ben Zaken et al., 2022) in NLP, we adopt Adapter (Houlsby et al., 2019) due to its simplicity. As shown in Fig. 2(a), Adapter is a bottleneck architecture which consists of two fully connected (FC) layers and an activation layer in the middle. The first FC layer projects the input to a lower dimension and the second FC layer projects it back to the original dimension. To adapt the pre-trained spatial features to target video data, we add an Adapter after the self-attention layer as shown in Fig. 2(c), which we term as spatial adaptation. During training, all the other layers of the transformer model are frozen while only the Adapters are updated. In Table 1, we show that our spatial adaptation strategy achieves comparable performance with the fully finetuned space-only baseline. This indicates that spatial adaptation helps the frozen image model to learn good spatial representations from video data. However, there is still a large gap between the performance of spatial adaptation and a fully finetuned video model because spatial adaptation alone lacks the ability to learn temporal information in videos.

## 3.3 TEMPORAL ADAPTATION

To capture temporal information more effectively, previous methods usually incorporate new temporal modules to pre-trained image models because it is commonly believed that image models cannot infer temporal structured information in videos. However, adding new temporal modules, either through temporal attention (Bertasius et al., 2021; Zhang et al., 2021b) or temporal encoder/decoder (Arnab et al., 2021; Lin et al., 2022), will introduce sizable number of extra tunable parameters. In addition, these new modules require full finetuning, which is inefficient.

To address this problem, we present a new strategy: *reuse the pre-trained self-attention layer in the image model to do temporal modeling.* More specifically, we denote the original self-attention layer as S-MSA for spatial modeling, and the reused self-atentnion layer as T-MSA for temporal modeling. As shown in Fig. 2(d), we put T-MSA in front of S-MSA. Now given the video patch embedding $z \in \mathbb{R}^{T \times (N+1) \times D}$, we first reshape it into $z^T \in \mathbb{R}^{(N+1) \times T \times D}$, where $N = HW/P^2$ is the number of spatial patches and $T$ is the number of frames. Then we feed $z^T$ into the T-MSA where it tries to learn the relationship among the $T$ frames. Note that T-MSA and S-MSA are the same layers (i.e., pre-trained MSA in the image model) and kept frozen during model tuning, but just applied to different input dimensions. This explicit operation helps our model with enhanced temporal modeling, while keeping the number of parameters fixed. In the end, similar to spatial adaptation, we add another Adapter after the reused temporal attention layer to adapt its features on video data, which we term as temporal adaptation. The structure of the Adapter is the same as in spatial adaptation but without the skip connection. The reason is that we want to initialize the adapted model to be close to the original model (Houlsby et al., 2019), thus we need to initialize the adapter to zero and remove the skip connection here to detach the effect of temporal adaptation at the beginning of training. As seen in Table 1, temporal adaptation helps to close the gap to fully finetuned video models while only introducing another lightweight Adapter into the transformer block.

## 3.4 JOINT ADAPTATION

Spatial and temporal adaptation are performed sequentially to different input dimensions with their individual purposes. It would be desirable to jointly tune the representations for spatiotemporal reasoning. To this end, we further introduce an Adapter in parallel to the MLP layer, which we term as joint adaptation. This Adapter has the same structure as the one in temporal adaptation.

The final structure of a transformer block in our proposed AIM is presented in Fig. 2(e). The computation of the adapted block can be written as

$$z_l^T = z_{l-1} + \text{Adapter}(\text{T-MSA}(\text{LN}(z_{l-1}))), \tag{3}$$

$$z_l^S = z_l^T + \text{Adapter}(\text{S-MSA}(\text{LN}(z_l^T))), \tag{4}$$

$$z_l = z_l^S + \text{MLP}(\text{LN}(z_l^S)) + s \cdot \text{Adapter}(\text{LN}(z_l^S)), \tag{5}$$

where $z_l^T$, $z_l^S$, $z_l$ denotes the temporal adapted, spatial adapted, and jointly adapted output in the $l$-th transformer block, respectively. Here, $s$ is a scaling factor to control the weight of the output from Adapter. For the final prediction, we simply take the average of the [class] tokens of each input frame and feed it to the classification head.

## 4 EXPERIMENTS

**Datasets.** We evaluate the proposed method on four widely adopted video action recognition benchmarks, Kinetics-400 (K400) (Kay et al., 2017), Kinetics-700 (K700) (Carreira et al., 2019), Something-something-v2 (SSv2) (Goyal et al., 2017) and Diving-48 (Li et al., 2018). K400 contains around 240K training videos and 20K validation videos in 400 human action classes. The videos are all trimmed to around 10 seconds. K700 is an extended version of K400 which contains around 530K training videos and 34K validation videos in 700 classes. SSv2 contains 168.9K training videos and 24.7K validation videos in 174 classes. SSv2 is more challenging because it requires stronger temporal modeling (Zhu et al., 2020; Sevilla-Lara et al., 2021). Diving-48 contains 15.9K training videos and 2K validation videos in 48 fine-grained diving actions. It is designed to be unbiased towards static representations, which means a model cannot simply rely on the objects or background to determine the action.

Table 1: Effectiveness of proposed components. We compare to three baselines on Something-something-v2 dataset. Spatial adaptation, temporal adaptation and joint adaptation gradually add spatiotemporal reasoning to the frozen image model. Views = #frames × #temporal × #spatial.

| Methods | Pretrain | Param (M) | Tunable Param (M) | Top-1 | Top-5 | Views |
|---|---|---|---|---|---|---|
| Frozen space-only | IN-21K | 86 | 0.1 | 15.1 | 36.9 | 8×1×3 |
| Finetuned space-only | IN-21K | 86 | 86 | 36.2 | 68.1 | 8×1×3 |
| Finetuned space-time (Bertasius et al., 2021) | IN-21K | 121 | 121 | 59.5 | 85.6 | 8×1×3 |
| Frozen space-only + spatial adaptation | IN-21K | 89 | 3.7 | 36.7 | 68.3 | 8×1×3 |
| + temporal adaptation | IN-21K | 97 | 10.8 | 61.2 | 87.7 | 8×1×3 |
| + joint adaptation (AIM) | IN-21K | 100 | 14.3 | **62.0** | 87.9 | 8×1×3 |
| AIM | CLIP | 100 | 14.3 | **66.4** | 90.5 | 8×1×3 |

## 4.1 EFFECTIVENESS OF COMPONENTS

To demonstrate the effectiveness of our proposed components in Sec. 3, we compare our method to three baselines. The first baseline is a frozen space-only model. Recall in Sec. 3.1, space-only model processes input frames independently and performs temporal average pooling in the end. We freeze the image backbone and only tune the classification head, which is also known as linear probing (He et al., 2020). The second baseline is a fully finetuned space-only model. It should be able to learn spatial information from video data, but still has difficulties in capturing temporal information. The third baseline is a fully finetuned space-time video model, which should serve as oracle. Here we choose TimeSformer (Bertasius et al., 2021) because we are based on the same ViT-B backbone and share a similar structure (*i.e.*, divided space-time attention).

In the experiments, we use the ViT-B/16 pre-trained on IN-21K as image backbone, and we compare the proposed method with the baselines on SSv2 (Goyal et al., 2017) where temporal modeling is critical. The results for three baselines are shown in Tab. 1 top. We can see that the frozen space-only model only needs to tune 0.1M parameters, but it also performs much worse than the fully finetuned video model (15.1% vs 59.5%). fully finetuning the space-only model allows it to learn improved spatial representations from video data and largely improves the performance (15.1% → 36.2%). However, it also significantly increases the number of tunable parameters and still has a large gap from the fully finetuned video model due to lack of temporal modeling. The third baseline, fully finetuned video model, achieves the highest accuracy due to its strong spatiotemporal reasoning capability, but the number of tunable parameters increases again to 121M.

Our goal is to add a few tunable parameters to the frozen space-only model and close the gap to fully finetuned video model. As shown in Tab. 1 bottom, after spatial adaptation, the frozen space-only model achieves comparable performance with the fully finetuned space-only model (36.7% vs 36.2%), with significantly less number of tunable parameters (3.7M vs 86M). This means spatial adaptation is able to help frozen image models to achieve good spatial modeling on video data. In addition, adding temporal adaptation further boosts the performance to 61.2%, which is even higher than the fully finetuned video model. This indicates that our temporal adaptation introduces strong temporal modeling to the space-only model. Finally, joint adaptation is incorporated to tune the features for improved spatiotemporal reasoning, which is our method AIM. We not only close the gap to fully finetuned space-time video model but obtain higher accuracy (62% vs 59.5%) with fewer number of tunable parameters (14.3M vs 86M). These results successfully validate the effectiveness of our proposed adaptation strategies.

Furthermore, our method could easily take advantage of stronger pre-trained image models and adapt them for video action recognition. For example, simply switch the ViT-B/16 pre-trained on IN-21K to CLIP pre-trained, we obtain another accuracy boost (62.0% → 66.4%)

## 4.2 COMPARISONS TO THE STATE OF THE ART

In this section, we compare the proposed method with state-of-the-art video models on four video action recognition benchmarks. For all the experiments, we use the ViT models pre-trained by CLIP (Radford et al., 2021). We mostly follow the training settings in Liu et al. (2022), and more implementation details can be found in Appendix.

Table 2: Comparison to state-of-the-art on Kinetics-400. Views = #frames $\times$ #temporal $\times$ #spatial.

| Methods | Pretrain | GFLOPs | Param (M) | Tunable Param (M) | Top-1 | Top-5 | Views |
|---|---|---|---|---|---|---|---|
| MViT-B (Fan et al., 2021) | - | 4095 | 37 | 37 | 81.2 | 95.1 | 64×3×3 |
| UniFormer-B (Li et al., 2021) | IN-1K | 3108 | 50 | 50 | 83.0 | 95.4 | 32×4×3 |
| TimeSformer-L (Bertasius et al., 2021) | IN-21K | 7140 | 121 | 121 | 80.7 | 94.7 | 64×1×3 |
| ViViT-L/16×2 FE (Arnab et al., 2021) | IN-21K | 3980 | 311 | 311 | 80.6 | 92.7 | 32×1×1 |
| VideoSwin-L (Liu et al., 2022) | IN-21K | 7248 | 197 | 197 | 83.1 | 95.9 | 32×4×3 |
| MViTv2-L (312 ↑) (Li et al., 2022) | IN-21K | 42420 | 218 | 218 | 86.1 | 97.0 | 32×3×5 |
| MTV-L (Yan et al., 2022) | JFT | 18050 | 876 | 876 | 84.3 | 96.3 | 32×4×3 |
| TokenLearner-L/10 (Ryoo et al., 2021) | JFT | 48912 | 450 | 450 | 85.4 | 96.3 | 64×4×3 |
| PromptCLIP A7 (Ju et al., 2021) | CLIP | - | - | - | 76.8 | 93.5 | 16×5×1 |
| ActionCLIP (Wang et al., 2021a) | CLIP | 16890 | 142 | 142 | 83.8 | 97.1 | 32×10×3 |
| X-CLIP-L/14 (Ni et al., 2022) | CLIP | 7890 | 420 | 420 | 87.1 | 97.6 | 8×4×3 |
| EVL ViT-L/14 (Lin et al., 2022) | CLIP | 8088 | 368 | 59 | 87.3 | - | 32×3×1 |
| AIM ViT-B/16 | CLIP | 606 | 97 | 11 | 83.9 | 96.3 | 8×3×1 |
| AIM ViT-B/16 | CLIP | 1214 | 97 | 11 | 84.5 | 96.6 | 16×3×1 |
| AIM ViT-B/16 | CLIP | 2428 | 97 | 11 | 84.9 | 96.7 | 32×3×1 |
| AIM ViT-L/14 | CLIP | 2802 | 341 | 38 | 86.8 | 97.2 | 8×3×1 |
| AIM ViT-L/14 | CLIP | 5604 | 341 | 38 | 87.3 | 97.6 | 16×3×1 |
| AIM ViT-L/14 | CLIP | 11208 | 341 | 38 | **87.5** | **97.7** | 32×3×1 |

### 4.2.1 RESULTS ON KINETICS-400 AND KINETICS-700

Tab. 2 presents the comparisons with state-of-the-art video models on K400 dataset. First, we can see that with ViT-B/16 backbone, our method only needs to tune 11M parameters for competitive performance, which is much smaller than previous video models. Taking input of 8 frames as an example, AIM ViT-B/16 achieves 83.9% top-1 accuracy while only requiring 606 GFLOPs. When using 16 input frames, our method even outperforms MTV-L (Yan et al., 2022), which requires more than 10× computations (1214 vs 18050 GFLOPs). When switching to larger backbone ViT-L/14, we achieve the highest accuracy 87.5% on K400 dataset, with 38M tunable parameters.

Note that several works also leverage CLIP pre-trained models to do video action recognition. However, ActionCLIP (Wang et al., 2021a) and X-CLIP (Ni et al., 2022) are multimodal methods which require additional text branch and tune the whole model end-to-end. PromptCLIP (Ju et al., 2021) applies prompt tuning (Lester et al., 2021) to CLIP and adds several temporal blocks for temporal modeling. EVL (Lin et al., 2022) introduces a new decoder branch to learn temporal information. However, AIM simply re-uses image pre-trained self-attention for temporal modeling. This makes AIM much simpler than previous methods, yet achieving better performance at much less tunable parameters. The simplicity also makes AIM much easier to adapt to different model architectures (single modal or multi-modal models). But previous methods such as ActionCLIP/X-CLIP/PromptCLIP cannot leverage pure image backbone because they need an additional text branch.

Furthermore, we evaluate our method on K700 dataset in Tab. 4. We can see that AIM ViT-B/16 with 11M tunable parameters is able to outperform MTV-L (875M) and MViTv2-B (51M). And AIM ViT-L/14 (38M) achieves comparable performance with MaskFeat (218M) (Wei et al., 2022). Note that MaskFeat uses larger input resolution (312 vs 224) and more input frames (40 vs 32) than us. This again justifies the effectiveness of our efficient adaptation pipeline.

### 4.2.2 RESULTS ON SOMETHING-SOMETHING-V2

Tab. 3 presents the performance comparisons on SSv2. Based on CLIP ViT-L/14, our method achieves competitive or better performance than most prior arts. In terms of fair comparison to EVL, which also uses CLIP pre-trained image encoder, we achieve significantly higher accuracy (70.6% > 66.7%), while introducing less tunable parameters (50M < 175M). Note that to introduce temporal modeling into image model, EVL adds 12 layers of decoder blocks, while our method simply reuse image pre-trained self-attention layers to achieve stronger temporal modeling .

However, our method falls behind some fully finetuned video models (Girdhar et al., 2022; Li et al., 2022; 2021). One reason is that SSv2 is a "temporal-heavy" dataset (Sevilla-Lara et al., 2021), which requires model to really understand the temporal evolution within a video. In order to obtain high accuracy, most previous video models are first pre-trained on some video datasets (such as

Table 3: Comparison to state-of-the-art on Something-Something-v2. K400[†]/K600[†] indicates the model is pre-trained on both IN-21K and K400/K600.

| Methods | Pretrain | GFLOPs | Param (M) | Tunable Param (M) | Top-1 | Top-5 | Views |
|---|---|---|---|---|---|---|---|
| TimeSformer-L (Bertasius et al., 2021) | IN-21K | 7140 | 121 | 121 | 62.4 | - | 64×1×3 |
| MTV-B (Yan et al., 2022) | IN-21K | 4790 | 310 | 310 | 67.6 | 90.4 | 32×4×3 |
| MViT-B (Fan et al., 2021) | K400 | 510 | 37 | 37 | 67.1 | 90.8 | 32×1×3 |
| MViTv2-B (Li et al., 2022) | K400 | 675 | 51 | 51 | 70.5 | 92.7 | 40×1×3 |
| ViViT-L/16×2 (Arnab et al., 2021) | K400[†] | 11892 | 311 | 311 | 65.4 | 89.8 | 16×4×3 |
| VideoSwin-B (Liu et al., 2022) | K400[†] | 963 | 89 | 89 | 69.6 | 92.7 | 32×1×1 |
| Omnivore (Girdhar et al., 2022) | K400[†] | - | - | - | 71.4 | 93.5 | 32×1×3 |
| MViTv2-L (312 ↑) (Li et al., 2022) | K400[†] | 8484 | 213 | 213 | **73.3** | **94.1** | 32×1×3 |
| UniFomer-B (Li et al., 2021) | K600[†] | 777 | 50 | 50 | 71.2 | 92.8 | 32×1×3 |
| CoVeR (Zhang et al., 2021a) | JFT-3B | - | - | - | 70.9 | - | - |
| EVL ViT-B/16 (Lin et al., 2022) | CLIP | 2047 | 182 | 86 | 62.4 | - | 32×1×3 |
| EVL ViT-L/14 Lin et al. (2022) | CLIP | 9641 | 484 | 175 | 66.7 | - | 32×1×3 |
| AIM ViT-B/16 | CLIP | 624 | 100 | 14 | 66.4 | 90.5 | 8×1×3 |
| AIM ViT-B/16 | CLIP | 1248 | 100 | 14 | 68.1 | 91.8 | 16×1×3 |
| AIM ViT-B/16 | CLIP | 2496 | 100 | 14 | 69.1 | 92.2 | 32×1×3 |
| AIM ViT-L/14 | CLIP | 2877 | 354 | 50 | 67.6 | 91.6 | 8×1×3 |
| AIM ViT-L/14 | CLIP | 5754 | 354 | 50 | 69.4 | 92.3 | 16×1×3 |
| AIM ViT-L/14 | CLIP | 11508 | 354 | 50 | 70.6 | 92.7 | 32×1×3 |

Table 4: Comparisons on Kinetics-700.

| Method | Pretrain | Tunable Param | Top-1 |
|---|---|---|---|
| VidTR-L (Zhang et al., 2021b) | IN-21K | 91 | 70.2 |
| MTV-L (Yan et al., 2022) | IN-21K | 876 | 75.2 |
| MViTv2-B (Li et al., 2022) | - | 51 | 76.6 |
| MViTv2-L (40 × 312 ↑) (Li et al., 2022) | IN-21K | 218 | 79.4 |
| MaskFeat (40 × 312 ↑) (Wei et al., 2022) | K700 | 218 | **80.4** |
| AIM ViT-B/16 | CLIP | 11 | 76.9 |
| AIM ViT-L/14 | CLIP | 38 | **80.4** |

Table 5: Comparisons on Diving-48.

| Method | Pretrain | Tunable Param | Top-1 |
|---|---|---|---|
| TimeSformer-L (Bertasius et al., 2021) | IN-21K | 121 | 81.0 |
| VideoSwin-B (Liu et al., 2022) | IN-21K | 88 | 81.9 |
| BEVT (Wang et al., 2022a) | K400[†] | 88 | 86.7 |
| SIFAR-B-14 (Fan et al., 2022) | IN-21K | 87 | 87.3 |
| ORViT (Herzig et al., 2022) | IN-21K | 160 | 88.0 |
| AIM ViT-B/16 | CLIP | 11 | 88.9 |
| AIM ViT-L/14 | CLIP | 38 | **90.6** |

K400/K600) to learn good spatiotemporal representations, then finetuned on SSv2. But our method still starts from the image pre-trained model. Another reason is that simply reusing the image pre-trained self-attention for temporal modeling may not be able to fully capture the complicated temporal information in SSv2 videos. This suggests that we need to conduct more temporal adaptation for these challenging "temporal-heavy" datasets.

### 4.2.3 RESULTS ON DIVING-48

A diving class in Diving-48 (Li et al., 2018) is defined by the combination of takeoff, movements in flight and entry, thus it requires the model to differentiate such fine-grained actions. As shown in Tab. 5, our method with 11M tunable parameters outperforms all prior methods. AIM ViT-L/14 further improves the top-1 accuracy to 90.6%. Comparing to ORViT (Herzig et al., 2022), despite they leverage additional object tracking model, our method still outperforms it with much less tunable parameters. This suggests that efficient finetuning can handle fine-grained action recognition.

## 5 DISCUSSION

**Different Pre-trained Models.** Here we demonstrate the effectiveness of AIM on different pre-trained models. In Table 6, we first show AIM based on ViT-B backbone. We compare AIM to TimeSformer because we use the same backbone (ViT-B) and have a similar structure (i.e., both using divided space-time attention). As can be seen, AIM achieves better performance than fully finetuned TimeSformer

Table 6: Performance of using different pre-trained models on K400.

| Model | Backbone | Pretrain | Tunable Param (M) | Mem (G) | Time (H) | Top-1 |
|---|---|---|---|---|---|---|
| TimeSformer | ViT-B | IN-21K | 121 | 10 | 20 | 78.5 |
| AIM | ViT-B | IN-21K | 11 | 7 | 15 | 78.8 |
| TimeSformer | ViT-B | CLIP | 121 | 10 | 20 | 82.0 |
| AIM | ViT-B | CLIP | 11 | 7 | 15 | 83.9 |
| VideoSwin-B | Swin-B | IN-21K | 88 | 18 | 64 | 82.7 |
| AIM | Swin-B | IN-21K | 9.2 | 9 | 37 | 82.1 |

under both IN-21K and CLIP pre-trained weights. Then we apply AIM to Swin-B backbone and compare it to VideoSwin when we both use Swin-B and IN-21K pre-training. Similarly, AIM achieves comparable performance with fully finetuned VideoSwin.

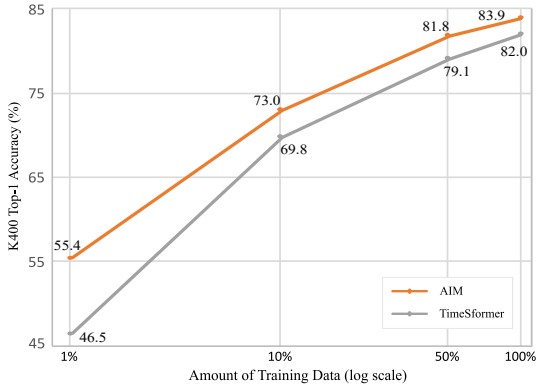

Figure 3: Data efficiency comparison. AIM outperforms fully finetuned TimeSformer under all scenarios, especially in low data regime.

Table 7: Effect of position of Adapters. Skip means adding Adapters every two blocks.

| Position | Tunable Param (M) | Top-1 |
|---|---|---|
| Bottom 6 | 5.6 | 80.7 |
| Top 6 | 5.6 | 83.3 |
| Skip | 5.6 | 83.2 |
| All | 11 | **83.9** |

Table 8: Effect of bottleneck ratio of Adapters.

| Ratio | Tunable Param (M) | Top-1 |
|---|---|---|
| 0.0625 | 3 | 83.3 |
| 0.125 | 5.6 | 83.4 |
| 0.25 | 11 | **83.9** |
| 0.5 | 21 | 83.8 |

**Data Efficiency.** One advantage of our efficient tuning paradigm is that we can keep the well pre-trained image representations intact. In the scenario where downstream data is insufficient, our method will be less prone to over-fitting compared to full finetuning. In Fig. 3, we compare AIM with fully finetuned TimeSformer under different amounts of training data on K400. For fair comparison, both AIM and TimeSformer use CLIP pre-trained ViT-B/16 as backbone. We can observe that under all scenarios, our method AIM outperforms fully finetuned TimeSformer. In particular, when the amount of data becomes less, the advantage of AIM becomes larger. For example, when there is only 1% of training data, we outperform TimeSformer by a significant margin of 8.9%.

**Training Cost.** Tab. 6 also shows the training time (hours) and memory cost (GB) of our method and fully finetuning on different backbones. All metrics are measured on 8 Tesla V100 GPUs. Compared to TimeSformer, we reduce the memory cost by 30% and training time by 25%. Compared to VideoSwin, we reduce the memory cost by 50% and training time by 42%.

**Position of Adapters.** By default, we add Adapters to every ViT block (12 blocks in total). Here we study the effect of adding Adapters in different layers. We add Adapters to the bottom 6 blocks (close to the input), top 6 blocks (close to the output) and one every two blocks. All these variants have the same number of tunable parameters. As can be seen in Tab. 7, adding Adapters to the bottom 6 blocks yields much worse performance than others. We hypothesize that the shallow layers learn generic representations which do not need much adaptation, while deeper layers learn task-specific features like temporal information thus feature adaptation is important. Adding Adapters to the top 6 blocks achieves comparable performance with adding to all blocks while saving half of the parameters. This could serve as a good candidate when training resources are more limited.

**Bottleneck Ratio of Adapters.** By tuning the bottleneck ratio of Adapters, we can easily control the number of tunable parameters. Here we study how the bottleneck ratio of Adapters affects the final performance. The results in Tab. 8 reveal that a larger bottleneck ratio tends to achieve better performance, but it will also introduce more tunable parameters. The performance plateaus after bottleneck ratio goes beyond 0.25. Note that a small ratio of 0.0625 could still achieve 83.3% top-1 accuracy on K400, which is competitive among state-of-the-art video models in Tab. 2 while introducing only 3M tunable parameters.

## 6 CONCLUSION

In this work, we propose a new way to efficiently transfer pre-trained image models for video action recognition. We introduce spatial adaptation, temporal adaptation and joint adaptation to gradually add spatiotemporal reasoning to an image model. Since only newly added Adapters are updated, our training cost is substantially lower than other fully finetuned video models. Yet we achieve comparable or even better performance than prior arts on four benchmarks. Our method is simple and generally applicable, which has the potential to leverage more powerful image foundation models in the future. Despite all the benefits, one limitation is that our simple strategy of reusing spatial attention for temporal modeling might not be strong enough for temporally challenging videos. Since video temporal modeling can be viewed as a form of sequence modeling, we might be able to reuse pre-trained weights from text or audio models instead of image models in the future.

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

# A  IMPLEMENTATION DETAILS

## A.1  KINETICS 400/700

We add spatial/temporal/joint adaptation in every ViT block as shown in Fig. 2. The bottleneck ratios of all adapters are set to 0.25 and the scaling factor is set to 0.5. The first FC layer in Adapters is randomly initialized and the second FC layer is initialized to zero. In this way, the adapted model is close to the pre-trained model at the beginning of training. We largely follow the training settings and data augmentations in Liu et al. (2022). Specifically, the model is trained for 30 epochs using AdamW (Kingma & Ba, 2014) optimizer with a batchsize of 64. The base learning rate is 3e-4 and weight decay is 5e-2. The learning rate is warmed up from 0 in the first 3 epochs and then decays following a cosine schedule. The stochastic depth rate is 0.2 for both ViT-B and ViT-L. For inference, we sample three clips along the temporal dimension. The final performance is evaluated by the ensemble of three views. We evaluate the model on 8, 16, 32 frames and the sampling interval is 16, 8, 4, respectively.

## A.2  SOMETHING-SOMETHING-V2

We add spatial/temporal/joint adaptation in every ViT block as shown in Fig. 2. We additionally add one adapter before T-MSA to enhance the temporal modeling. The bottleneck ratios of all adapters are set to 0.25 and the scaling factor is set to 0.5. We follow Liu et al. (2022) to use stronger data augmentations including label smoothing, RandAugment (Cubuk et al., 2020) and random erasing (Zhong et al., 2020). The model is trained for 50 epochs using AdamW (Kingma & Ba, 2014) optimizer. The other training settings are the same as Kinetics-400. We uniformly sample 8, 16, 32 frames in the experiments. For inference, we sample three spatial crops. The final performance is evaluated by the ensemble of three views.

## A.3  DIVING-48

We add spatial/temporal/joint adaptation in every ViT block as shown in Fig. 2. The bottleneck ratios of all adapters are set to 0.25 and the scaling factor is set to 0.5. The model is trained for 50 epochs. The other training settings and data augmentations are the same as K400. We uniformly sample 8, 16, 32 frames in the experiments. For inference, we only sample 1 temporal clip.

# B  VISUALIZATION

In this section, we present the attention map visualizations of the frozen space-only model, Spatial Adaptation (SA) model, Spatial Adaptation plus Temporal Adaptation (TA) model, and the fully finetuned TimeSformer.

On Fig. 4 left, we visualize an action "Brush Painting" from Kinetics-400 dataset. We can see that the attention maps of the frozen space-only model are very scattered, and it doesn't attend to the brush region in the first two frames. Adding SA enhances the attention on the brush, but the model still focuses on areas that are unrelated to the action. Further adding TA helps the model to learn temporal information. We can see that the model now focuses more on the brush painting area, which is similar to what fully finetuned TimeSformer does.

On Fig. 4 right, we visualize an action "Something falling like a rock" from Something-Something-v2 dataset. To correctly recognize this action, the model needs to learn how the object moves in the input frames. We first observe that both the frozen space-only model and SA model have good attention on the object, but they fail to model the movement of the object which leads to wrong prediction. In contrast, TA helps the model to learn the relationship among input frames. The attention map shows that the model not only focuses on the object but also learns the track of the object. Instead, TimeSformer always attends to the bottom region without showing the object path.

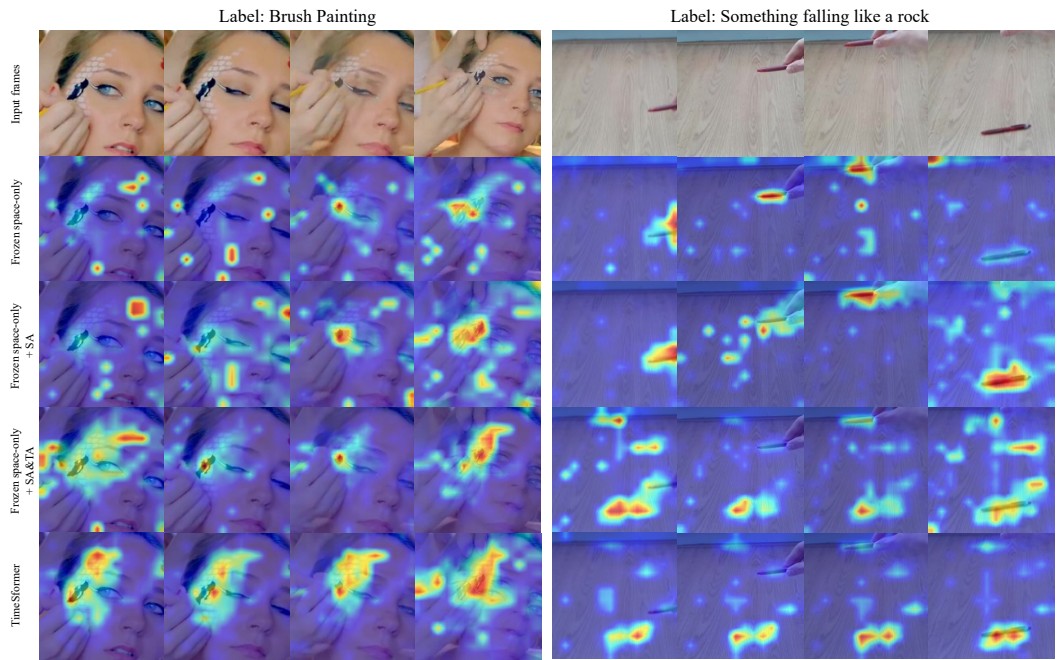

Figure 4: Attention map visualizations of AIM variants and the fully finetuned TimeSformer. With the help of temporal adaptation (TA), our method is able to focus on motion salient regions which helps to make a correct prediction.

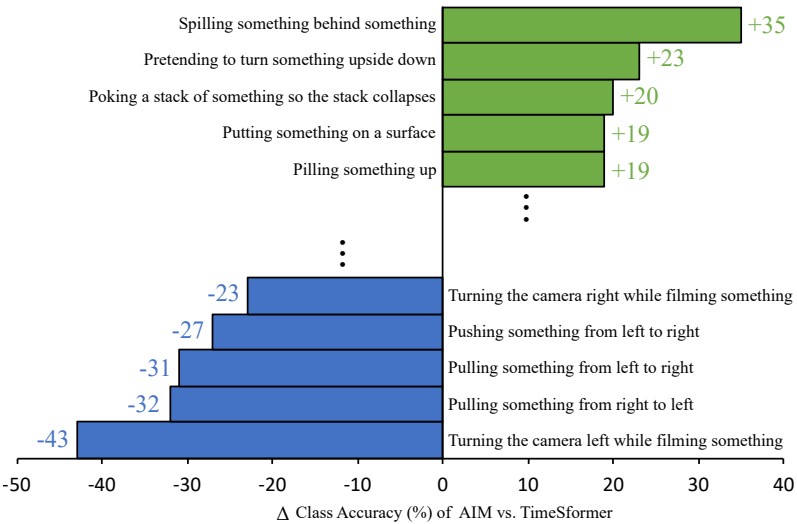

Figure 5: The figure shows the differences of each class's accuracy of AIM and TimeSformer on Something-Something-v2. Here we only plot the top-5 and bottom-5 classes.

## C  PER-CLASS ANALYSIS

In Tab. 3, we show that AIM still falls behind some SoTA fully finetuned video models on the "temporal-heavy" Something-Something-v2 (SSv2) dataset. We conjecture one reason is that simply reusing the image pre-trained self-attention for temporal modeling may not be able to fully capture the complicated temporal information in some nuanced action classes in SSv2. To provide further insights, we compute the per-class accuracy differences of AIM and TimeSformer on SSv2 and show the top-5 and bottom-5 classes in Fig. 5. We can see that the classes where AIM performs better are normal action classes with decent motion. The classes where AIM performs worse are those with

Table 9: Comparisons of the training memory cost of AIM and fully finetuned models based on large image pre-trained backbones. AIM significantly reduces the memory cost and makes large model training easier.

| Model | Backbone | Mem (G) |
|---|---|---|
| TimeSformer Bertasius et al. (2021) | ViT-L | 21.2 |
| AIM | ViT-L | 14.3 |
| VideoSwin Liu et al. (2022) | Swin-L | Out of Memory |
| AIM | Swin-L | 13.7 |

Table 10: Comparisons with EVL under different pre-trained datasets. AIM outperforms EVL under different pre-training and uses less number of tunable parameters.

| Model | Backbone | Pretrain | Tunable Param (M) | Mem (G) | Time (H) | Top-1 |
|---|---|---|---|---|---|---|
| EVL Lin et al. (2022) | ViT-B | IN-21K | 36.3 | 4.2 | 29 | 75.4 |
| AIM | ViT-B | IN-21K | 11 | 7 | 15 | **78.8** |
| EVL Lin et al. (2022) | ViT-B | CLIP | 36.3 | 4.2 | 29 | 82.9 |
| AIM | ViT-B | CLIP | 11 | 7 | 15 | **83.9** |

minor differences (*e.g.*, "Pulling something from left to right" vs. "Pulling something from right to left"). In order to tell these actions apart, the model needs to distinguish between the nuances, especially in motion. Given most of model parameters are frozen in our method, AIM may lack the capacity to capture such complex temporal information.

## D   MORE COMPARISONS OF TRAINING COST

In Table 6, we demonstrate the training efficiency of AIM based on ViT-B and Swin-B backbones. In this section, we show more comparisons with fully finetuned baselines based on ViT-L and Swin-L backbones. The results are shown in Table 9. We can see that TimeSformer with a ViT-L backbone needs 21.2G GPU memory, and VidesSwin with a Swin-L backbone cannot fit into an 8 Tesla V100 32G GPU server. In both cases, AIM can significantly reduce the memory usage to 14.3G and 13.7G, respectively. This makes large model training more memory-friendly (runnable on most GPUs with 15G memory and more) , and thus more affordable for most researchers and practioners.

Furthermore, beyond memory saving, optimizing number of tunable parameters has potential benefits in other applications such as communication-efficient distributed learning (e.g., federated learning where the tunable model parameters are communicated between the central server and local clients) and privacy preserving federated learning Zhao et al. (2022); Yu et al. (2021). Tuning less parameters could also be beneficial when the downstream data is limited because fully finetuning a large model on limited data may suffer from serious overfitting. This can be observed from the results in Fig. 3 where AIM obtains larger accuracy improvements over the fully finetuned baseline when there is only small amount of training data.

## E   COMPARISON TO EVL UNDER DIFFERENT PRE-TRAINED DATASETS

In this section, we compare AIM with EVL Lin et al. (2022), which is the most recent SoTA image-to-video efficient finetuning method based on frozen pre-trained ViT. As shown in the Table 10, AIM consistently outperforms EVL under both IN-21K and CLIP pre-training as well. And AIM uses considerably less tunable number of parameters than EVL.

## F   PSEUDO-CODE OF THE ADAPTED VIT BLOCK

As explained in the paper, AIM is effective and simple to implement. In Algorithm 1, we show the PyTorch style pseudo-code on how to apply AIM to a ViT block.

**Algorithm 1** Pseudo-code of an adapted ViT block

```python
class TransformerBlock():

    def __init__(self, dim, num_head, mlp_ratio, scale):
        ## Layers in the original ViT block
        self.attn = MultiheadAttention(dim, num_head)
        self.norm1 = LayerNorm(dim)
        self.mlp = MLP(dim, mlp_ratio)
        self.norm2 = LayerNorm(dim)

        ## Adapters
        self.s_adapter = Adapter(dim)
        self.t_adapter = Adapter(dim)
        self.mlp_adapter = Adapter(dim)
        self.scale = scale

    def forward(x):
        ## x in shape [N+1, BT, D]

        ## temporal adaptation
        xt = rearrange(x, 'n (b t) d -> t (b n) d', t=num_frames)
        xt = self.t_adapter(self.attn(self.norm1(x)))
        xt = rearrange(x, 't (b n) d -> n (b t) d', n=num_patches)
        x = x + xt

        ## spatial adaptation
        x = x + self.s_adapter(self.attn(self.norm1(x)))

        ## joint adaptation
        x_norm = self.norm2(x)
        x = x + self.mlp(x_norm) + self.scale * self.mlp_adapter(x_norm)

        return x
```

