# OpenReview forum: "AIM: Adapting Image Models for Efficient Video Action Recognition"
_ICLR.cc/2023/Conference — ICLR 2023 poster_

### Official Review · Reviewer_auyL · 2022-10-23

**Confidence:** 4
**Correctness:** 3
**Technical Novelty And Significance:** 3
**Empirical Novelty And Significance:** 3
**Recommendation:** 6

**Clarity, Quality, Novelty And Reproducibility:**

Overall, the paper is well written and easy to follow. However, the experiments are missing some important ablations (e.g. the comparison with traditional full tuning scheme, the baselines with same pre-training dataset, etc.) and this make the paper not solid enough.

**Strength And Weaknesses:**

Strength:
1. The paper is well motivated. I agree that the well pre-trained model weights are not necessary to be fine-tuned for adapting to video tasks.
2. The paper is well written and easy to follow.
3. The proposed methods are parameter efficient based on the presented experiments.
4. With the adapters inserted, even with less number of tunable parameters and memory cost, AIM achieves higher accuracy than ViT baseline method. Could the authors explain why this could happen?

Weakness:
1. The efficiency is not clear in terms of memory consumption, latency and throughput. The discussion on the saving on the memory and training hours are reported separately in Tab. 6 in the paper. I would suggest to put the memory and latency metrics in Tab. 3 also to better reflect the advantages over baseline methods.
2. The ablations on the impact of AIM over traditional full tuning is not solid. It would be clearer to see the impact of adapter by comparing the AIM to traditional full tuning method. Currently, most of the method selected in Tab.3, Tab. 4 and Tab. 5 are not pre-trained on CLIP, while AIM methods are all pre-trained on CLIP. Then, it is hard to check, how much benefits come from AIM and how much of the benefits come from CLIP pre-training. Although this is explored separately in Tab. 6, the comparison with baselines models are missing.



**Summary Of The Paper:**

The paper proposes an efficient method of transferring image pre-trained weights to video processing. In summary, the paper proposes to freeze most of the pre-trained model parameters with only a small portion of parameters in the newly inserted Adapter module learnable. The authors have show the effectiveness of the proposed AIM methods in several datasets and significant improvements over the selected baselines.

**Summary Of The Review:**

Overall, the paper is well motivated and the proposed solution is reasonable. However, due to some missing ablations, I keep my score as boardline (leaning accept). I will change my score accordingly based on the author's response.

---

### Official Review · Reviewer_UEFx · 2022-10-23

**Confidence:** 4
**Correctness:** 4
**Technical Novelty And Significance:** 2
**Empirical Novelty And Significance:** 2
**Recommendation:** 6

**Clarity, Quality, Novelty And Reproducibility:**

## Quality [medium]
The paper quality is good. The method is well motivated and properly evaluated. It would have been stronger if it were evaluated on more diverse video tasks like detection/question-answering etc.

## Reproducibility [medium]
The approach is fairly simple and the appendix has enough information to repro the method, so I don't have any major concerns. One issue with video datasets like kinetics is that the available data keeps changing, so authors should release the kinetics video IDs used in training and testing for fair comparisons. Although since results on SS-v2 are provided so those results at least should be easily reproducible.

## Novelty [low]
I think this is the major limitation of the work, as noted in weaknesses. The proposed approach is fairly straightforward application of the adapter idea to images->videos.

## Clarity [high]
The paper is generally well written and barring some small issues, is as an easy read. Implementation details are provided. I note some minor issues here:
- pg 2: "Particularly we use the mostly studied video action recognition as an illustrating task." --> rephrase
- pg 8: "resources is more limited" --> "resources are more limited"

**Strength And Weaknesses:**

## Strengths
1. Simple and effective approach and doesn't need the text trunk: Perhaps the strongest part about this paper is the simplicity of the approach. Even compared to recent work like EVL (Lin et al. 2022), which trains a decoder to combine frames, and ActionCLIP/XCLIP/CLIP-ViP ((https://arxiv.org/abs/2209.06430) which leverage the text trunk as well, this method simply uses the visual trunk only and is able to adapt it for supervised classification training without needing any text prompting. It simply leverages the adapter architecture proposed in NLP ((Houlsby et al., 2019).

2. Shows gains in data efficient setup: The proposed approach seems particularly effective in few-shot settings. It is however unclear how it compares to few-shot performance of the other contemporary CLIP->video methods (as in point 1 above).

## Weaknesses
1. Somewhat weak results: As noted in strength point 1, CLIP->video models is a very competitive space. Although the method is simpler, the performance isn't particularly better than other approaches as noted in most of the tables. Also the performance does lag fully supervised end-to-end trained methods like Omnivore and MVIT-v2 on challenging temporal reasoning datasets like SSv2 (Table 3), (Omnivore/MVIT don't even use CLIP features). This suggests the approach may not scale well towards challenging temporal reasoning video tasks and is probably best suited for Kinetics-like tasks (which anyway works well even with the original frame-level CLIP features).

2. Compute savings aren't huge: Another analysis/aspect I liked about the method was compute analysis in Table 6. However as the numbers show, the huge gains in parameter savings don't really transfer to compute/train-time/memory savings as much. Since the trainable parameter count isn't a particularly useful metric to optimize for, it's not clear if users/practitioners are indeed better off with adding these additional adapter layers, or should they just finetune the full model without modifying the architecture at a bit higher runtime cost (30% in case of ViT), but a near guarantee of better performance.

**Summary Of The Paper:**

The paper proposes a simple way for adapting image pretrained models to videos. Specifically, authors leverage the default ViT architecture, add a temporal attention layer simply by using the same weights learned for spatial attention, and add lightweight `adapter' layers that consists of a bottleneck block. Only the adapter layers are trained, and they are added after the spatial attention, temporal attention, and as a residual to the last frozen MLP in the transformer block. The resulting model obtains strong results on 4 benchmark, closely matching SOTA. The resulting model is also data efficient and gets larger gains with small amounts of training data (Fig 3).

**Summary Of The Review:**

I am very borderline on this paper. I'm leaning towards accept since this is a useful technique (initializing from image models), solves an important problem (video recognition), and is perhaps the simplest instantiation of a solution.

---

### Official Review · Reviewer_jsrf · 2022-10-24

**Confidence:** 4
**Correctness:** 3
**Technical Novelty And Significance:** 3
**Empirical Novelty And Significance:** 4
**Recommendation:** 8

**Clarity, Quality, Novelty And Reproducibility:**

- I believe the paper is reproducible , but still expect the author(s) release code/models upon publication.
- The paper quality, clarity, novelty are in good shape and ready for publication. This method will be useful for academic labs with limited hardware/computational resources, AIM enables them to train videos with budget resources and still obtain SoTA performance.

**Details Of Ethics Concerns:**

I do not foresee any ethics concerns.

**Strength And Weaknesses:**

# Strength
- The idea of adapting image models (pretrained on large benchmarks) to video classification with minimum numbers of tunable parameters and training time is novel and interesting (not truly novel due to Lin et al. 2022, but the authors clearly state the differences).
- AIM showed many advantages compared with SoTA methods where full-fine-tuning is used. AIM is particularly efficient in terms of number of tunable parameters, training time, and data efficiency. In most cases, we observe AIM provides better performances (except for SSv2).
- Ablation experiments are enough to understand the design choice.
- The trick that using weight from spatial-attention for temporal attention (then keep them frozen) is nice. In deep learning we saw this trick many times (e.g., CDC or inflation of image models) but most of the time, it is followed by full-fine-tuning. AIM re-purpose the spatial weights for temporal task, w/o full-fine-tuning, instead adapting them. This makes it quite interesting.

# Weakness
- Not a major weakness, but the paper titled as "video understanding", it is more suitable to make it "video classification" or "video recognition" since all benchmarks and tasks experimented in the paper are classification.

**Summary Of The Paper:**

This paper proposes AIM (Adapting Image Models) for video classification with the focus on reducing tunable parameter and training (fine-tuning) time. The main contributions are: 1) using Adapter (Houlsby et al. 2019) to adapt image models (pretrained on large datasets such as ImageNet-21k or CLIP); 2) designing various spatial, temporal, and joint adaptation blocks. The paper provides strong ablation experiments and solid results compared with state-of-the-art (SoTA) methods on multiple benchmarks including K-400, K700, SSv2, and Diving-48. Written presentation is clear and easy to read.

**Summary Of The Review:**

In summary, the paper is novel and interesting, experiments are solid and potential impactful (useful for academic labs). I recommend to accept this paper.

---

### Decision · Program_Chairs · 2023-01-20

**Decision:**

Accept: poster

**Justification For Why Not Higher Score:**

- Empirical results are solid, but not amazing.

**Justification For Why Not Lower Score:**

- Solid result in an important area of video understanding
- Potential for wide adoption

**Metareview: Summary, Strengths And Weaknesses:**

The authors propose a method for adapting pretrained image models (AIM) to video classification by freezing most of the pre-trained parameters and training only specific adapter modules. The authors consider various spatial, temporal, and joint adaptation blocks and evaluate the results on multiple benchmarks (K400, K700, SSv2, and others) and show competitive performance with a significant saving in terms of the number of trainable parameters. The reviewers agreed that this work is relevant for the larger community and has potential for wider adoption given the strong empirical results and the simplicity of the proposed method.

**Note From Pc:**

if the above contains the word "oral" or "spotlight" please see: "oral" presentation means -> notable-top-5% and "spotlight" means -> notable-top-25%. As stated in our emails, we are disassociating presentation type from AC recommendations